# A Novel Query-Driven Multi-Stage Alternating Feature Extraction and Interaction Network for Image Manipulation Localization

## Abstract

Image Manipulation Localization (IML) aims to identify and localize the tampered regions within edited images. Many studies employ a dual-branch backbone to extract tampering features from dual modalities, followed by feature fusion at the final stage. In this process, the extraction and fusion of dual-modality features is relatively independent, which fails to fully leverage the complementarity between different modalities and thus diminishes sensitivity to tampering artifacts. Inspired by the way humans continuously integrate multi-faceted knowledge to understand the world, we propose QMA-Net, which contains a novel Multi-stage Alternating Feature Extraction and Interaction architecture. At each stage, we deeply explore the intrinsic relationships and mappings between different modality features. Feature extraction and interaction are performed alternately, constructing complementary dual-modality tampering feature representations and enhancing sensitivity to tampering artifacts. Additionally, we introduce a lightweight, Query-driven Multi-level Feature Decoding. This mechanism progressively aggregates key information from multi-level dual-modality tampering features through multiple sets of learnable tamper-aware queries, effectively filtering out irrelevant features. Finally, multi-level queries are used to refine discriminative features, enabling precise localization of tampered regions. Extensive experiments demonstrate that our framework outperforms current state-of-the-art models in localization accuracy and robustness across multiple public datasets, achieving a favorable balance between performance and efficiency.

## 1 Introduction

The widespread use of digital image tampering techniques poses a severe challenge to societal trust systems. Tampered images are often used to create fake news, forge evidence, or commit fraud, posing serious threats to personal privacy, social order, and even national security. IML has become a critical technological barrier for maintaining information authenticity.

Most existing IML frameworks (Zhu et al., 2025)(Zeng et al., 2024)(Guo et al., 2024)(Lin et al., 2023) typically adopt a dual-branch structure, where one branch extracts features from the RGB modality, and the other extracts features from noise or high-frequency modality, working together to locate tampered regions. These frameworks often follow the classical paradigm of "extraction → simple fusion". This paradigm suffers from severe modality isolation and information fragmentation, which results in a lack or insufficiency of interaction among features at early and intermediate levels. For example, the RGB modality focuses on semantic boundaries but cannot perceive local statistical anomalies present in the noise modality; conversely, the noise modality is sensitive to compression artifacts but lacks high-level semantic guidance, often misclassifying highly textured regions (e.g., leaves) as tampered. Therefore, we hope to design a comprehensive bidirectional inter-modal interaction mechanism to construct complementary dual-modality tampering feature representations, thereby enhancing the sensitivity of dual-modality features to tampering artifacts. Through experiments, we visualize the features output by the backbone under both classical paradigms (Columns 5 and 6) and our methods (Columns 3 and 4) in Fig 1. It can be observed that dual-modality features under our methods exhibit higher sensitivity to tampered regions.

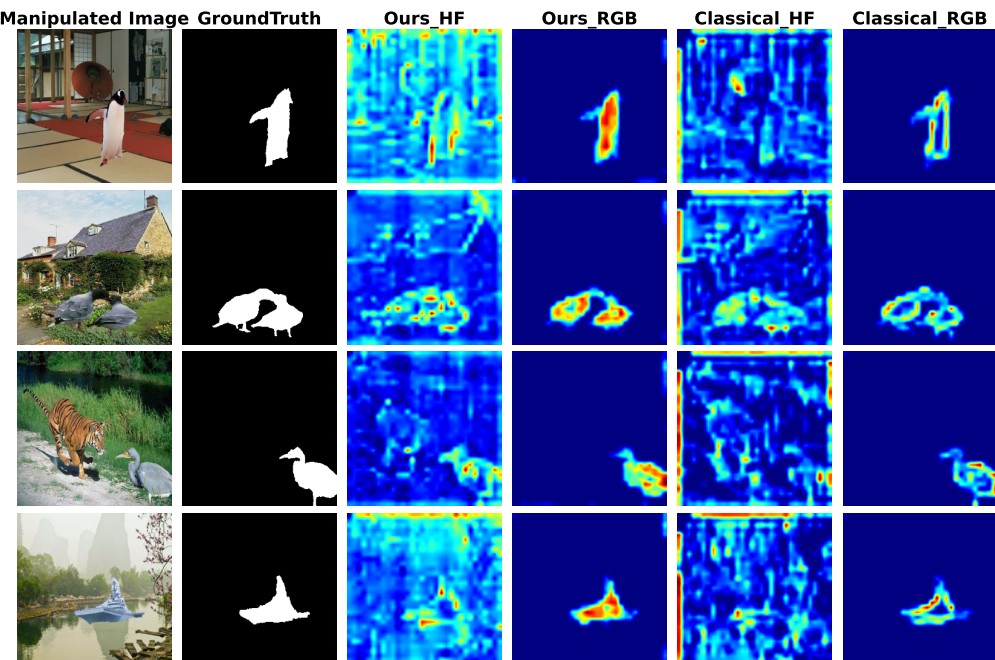

Figure 1: Ours vs. Classical. Grad-CAM visualizations of backbone output features under different paradigms. The redder the color of the region, the greater its contribution to the model's prediction results. "HF" denotes the high-frequency noise modality.

Current multi-level feature decoding methods often employ upsampling combined with convolution or MLP (Zhuang et al., 2021)(Ma et al., 2023), which indiscriminately aggregate a large amount of irrelevant background information and noise, severely overwhelming subtle tampering features. Moreover, fixed or simple decoding strategies struggle to adaptively balance the contributions of features from different levels and modalities, preventing effective collaboration between shallow fine-grained textures and deep semantic context, and thus degrading localization accuracy. In addition, computationally intensive decoding hinders their deployment in resource-constrained scenarios. Therefore, we decide to design the decoding mechanism of our framework to be lightweight and capable of effectively screening and aggregating critical information from multi-level features.

Recent studies have introduced multimodal large models into the task of IML Yin et al. (2024)(Zhang et al., 2024)(Su et al., 2024)(Kwon et al., 2025), using them as a backbone to extract general features. While this approach enables the extraction of more comprehensive tampering features, it also introduces a large amount of irrelevant information (Zhang et al., 2025a). Without sufficient feature selection, this can easily lead to counterproductive results. Moreover, multimodal large models have more critical limitations, including the need for substantial computational resources and slower inference speed, making deployment on mobile devices challenging. Our model aims to incorporate as few additional parameters as possible based on a relatively lightweight backbone, achieving a favorable balance between performance and efficiency.

Motivated by the above observations, we propose QMA-Net, which contains a Multi-stage Alternating Feature Extraction and Interaction architecture and a lightweight, Query-driven Multi-level Feature Decoding. The former is inspired by the human cognitive process of repeatedly examining complex objects from multiple perspectives to gain a deep understanding. At each stage, feature extraction and deep interaction are not isolated but performed alternately and mutually reinforced. Through specially designed cross-modal Feature Alignment and Dual-modal Feature Cross-guided Module, the network bidirectionally and deeply explores previously underutilized intrinsic relationships and mappings between RGB and noise modality features at every stage. This process essentially enables both modalities to perform bidirectional retrieval and attention across multiple levels, perceiving important relevant points within each modality while suppressing irrelevant noise. To allow such interaction to propagate across levels, the interacted features are fed back into the

backbone to extract the next-level features. In this way, we successfully construct complementary dual-modality tampering feature representations, significantly enhancing sensitivity to tampering artifacts. Furthermore, we innovatively introduce a lightweight decoding mechanism. By incorporating multiple sets of learnable, dedicated tamper-aware query vectors, these queries act as "fusion controller" for each feature level. At each stage, through a carefully designed Multi-domain Feature Aggregation Module, the queries progressively and selectively extract and condense the most critical information from dual-modality features while filtering out irrelevant interference. This process effectively simulates a coarse-to-fine, stepwise focusing decision procedure: shallow-level queries target potential anomalous regions, whereas deep-level queries associate global context to verify and refine localization results. Finally, the multi-level dual-modality features and their corresponding queries are fed into a Query-driven Multi-level Feature Decoder to localize the tampered regions. Such a dynamic, decision-level fusion strategy substantially improves localization accuracy.

In summary, our main contributions are as follows:

- We introduce a novel framework, QMA-Net, which contains a Multi-stage Alternating Feature Extraction and Interaction architecture. Feature extraction and interaction are performed alternately. At each stage, cross-modal Feature Alignment and Dual-modal Feature Cross-guided Module are employed to deeply explore intrinsic relationships and mappings between different modality features. This approach constructs complementary dual-modality tampering feature representations and significantly enhances sensitivity to tampering artifacts.

- We propose a lightweight, Query-driven Multi-level Feature Decoding. We introduce multiple sets of learnable tamper-aware queries, which progressively aggregate key information from dual-modality tampering features at each stage through a Multi-domain Feature Aggregation Module, while filtering out irrelevant features. In the Query-driven Multi-level Feature Decoder, these queries act as "fusion controllers", performing decision-level selection and fusion of dual-modality features at each level, substantially improving localization accuracy.

- Extensive experiments demonstrate that our framework outperforms existing state-of-the-art models in both localization accuracy and robustness across multiple public datasets. Moreover, our framework has a relatively small number of parameters and low computational requirements, making it more suitable for practical applications.

## 2 RELATED WORKS

### 2.1 IMAGE MANIPULATION LOCALIZATION

Traditional IML methods mainly rely on hand-crafted extractors (Ferrara et al., 2012)(Ye et al., 2007)(Tralic et al., 2012)(Kong et al., 2025) to capture anomalies caused by tampering operations. For example, Pasquale et al. utilize CFA (Ferrara et al., 2012) pattern inconsistencies to detect forged regions. However, they rely on the assumption that specific tampering operations always leave particular traces. When this assumption is not valid, their performance deteriorates significantly. With the development of computational power, deep learning has achieved remarkable progress in the field of IML (Cozzolino & Verdoliva, 2020)(Bappy et al., 2019)(Hao et al., 2021)(Hu et al., 2020). Wu et al. propose ManTra-Net(Wu et al., 2019), which formulates IML as an anomaly detection problem by modeling 385 manipulation types with Z-score and auxiliary features. Liu et al. introduce PSCC-Net(Liu et al., 2022), which leverages HRNet(Wang et al., 2021) to learn multi-scale features and employs SCCM to capture spatial–channel correlations for progressive mask generation. ObjectFormer (Wang et al., 2022) employs CNN layers to extract local features and then leverages transformer encoders for global modeling. However, they still exhibit limitations in generalization capability and robustness. In the latest studies, researchers have begun exploring multimodal large models (MLM) in IML to improve generalization and interpretability. For instance, FakeShield (Xu et al., 2025) utilizes MLM by integrating visual features with linguistic instructions, enabling instruction-driven forgery localization and interpretable outputs. IMDPrompter (Zhang et al., 2025b) leverages a Cross-view Prompt Learning paradigm built upon SAM(Kirillov et al., 2023) to achieve robust localization. Nevertheless, these approaches often come with higher computational demands, slower inference speed, and introduce irrelevant features.

## 2.2 CLASSICAL PARADIGM OF FEATURE EXTRACTION AND FUSION IN IML

The dual-branch architecture that separately extracts RGB features and high-frequency modality features has become a classic paradigm for IML. This paradigm typically adopts a sequential feature extraction and simple fusion strategy. For example, MVSS-Net (Dong et al., 2023) performs feature extraction and fusion in an independent manner, where only the deepest features from ResNet (He et al., 2016) are used for fusion, while shallow features are largely ignored. Such a design limits the model's ability to capture fine-grained cues and hinders comprehensive representation learning. (Mazumdar & Bora, 2022) propose the two-stream encoder–decoder network, where two branches independently perform encoding and decoding without cross-branch interaction, thus limiting complementary feature learning. (Niu et al., 2024) propose an end-to-end IML network that fuses RGB and noise features at each level using FAM, but without feeding the fused features back to the backbone, which restricts the transmission of fusion information across levels. In summary, such approaches lead to suboptimal representation learning.

## 2.3 QUERY-BASED FEATURE AGGREGATION

Query-based architectures have recently emerged as a powerful paradigm for feature selection in computer vision. The fundamental principle lies in introducing learnable queries that interact with image features through attention mechanisms, thereby steering the network toward task-relevant regions. For example, Mask2Former (Cheng et al., 2022) employs masked cross-attention to refine queries, restricting attention to predicted mask regions and thereby improving segmentation precision. ECENet (Liu et al., 2023) further generates object queries directly from predicted masks, ensuring that each query is semantically explicit and corresponds to a distinct object region. In SAM (Kirillov et al., 2023), output tokens serve as queries that guide image embeddings and prompt embeddings for segmentation. These methods demonstrate the outstanding potential of learnable queries in feature selection and aggregation.

## 3 METHODOLOGY

### 3.1 OVERALL FRAMEWORK OF QMA-NET

We propose an IML framework, QMA-Net (as shown in Fig 2), which emulates human learning mechanisms to acquire feature representations sensitive to tampering artifacts. It is primarily composed of two key components: a Multi-stage Alternating Feature Extraction and Interaction, and a Query-driven Multi-level Feature Decoding. The former employs both the RGB view $X_{img}$ and a high-frequency noise view $X_{noise}$ processed by a Multi-domain Noise-sensitive Fusion Module (MNFM) as inputs. The interrelationships between diverse modal features are extensively explored at each stage via a cross-modal Feature Alignment and Dual-modal Feature Cross-guided Module (DFCM). The enhanced features are subsequently fed into the corresponding backbone for next-level feature extraction. This process facilitates the construction of complementary dual-modal feature representations $\{R_1 \sim R_4, N_1 \sim N_4\}$, thereby improving sensitivity to tampering artifacts. The latter employs a set of learnable tamper-aware queries $Q_{taq}$ as input. At each stage, key information from dual-modal tampering features is progressively integrated through a Multi-domain Feature Aggregation Module (MFAM), while irrelevant features are filtered out, resulting in multi-level query embeddings $\{Q_1 \sim Q_4\}$. Finally, the dual-modal features and their corresponding query embeddings at each level form a triplet, which is fed into the Query-driven Multi-level Feature Decoder (QMFD) for decoding, enabling precise localization of tampered regions.

### 3.2 MULTI-STAGE ALTERNATING FEATURE EXTRACTION AND INTERACTION

Previous methods (Zeng et al., 2024)(Guo et al., 2024) employ a simple approach for feature extraction and fusion. These methods fail to achieve sufficient feature interaction and do not effectively leverage the complementarity between dual-modal features. Accordingly, we propose a Multi-stage Alternating Feature Extraction and Interaction. It follows a continuously "extract→interact→extract→interact" paradigm, where feature extraction and interaction are performed alternately. SegFormer (Xie et al., 2021) is selected to serve as both the Global and Local Artifact Extractors (GAE and LAE) in the network, and the architecture is manually divided into

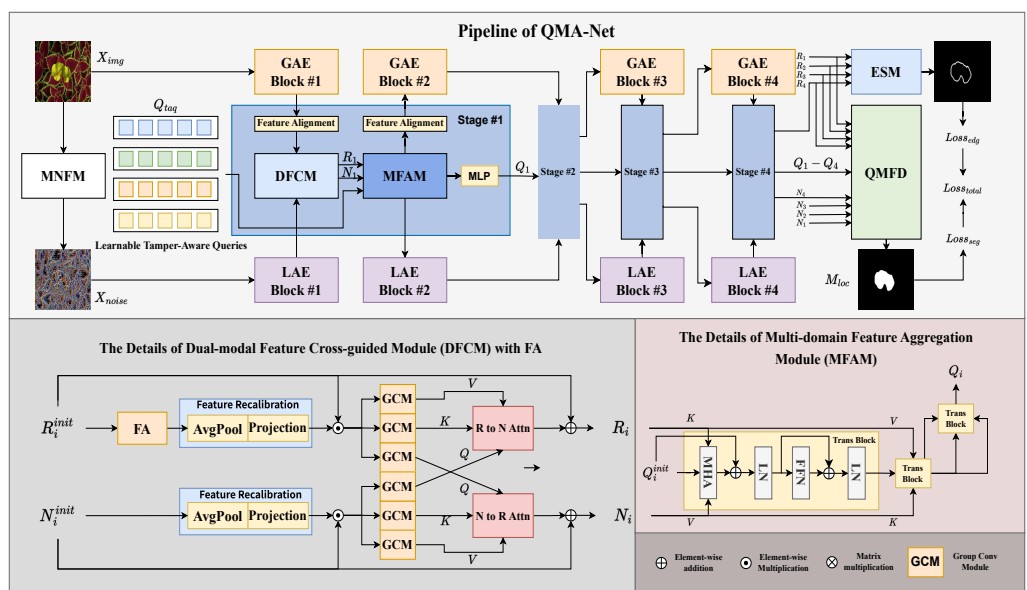

Figure 2: The pipeline of QMA-Net. Given an RGB view $X_{img}$ and a high-frequency noise view $X_{noise}$, they are first processed by the Multi-stage Alternating Feature Extraction and Interaction architecture to generate multi-level complementary feature representations $\{R_1 \sim R_4, N_1 \sim N_4\}$. Subsequently, a set of learnable tamper-aware queries $Q_{taq}$ progressively aggregates cross-modal information through the MFAM. Finally, the tampering features and query embeddings from all levels are fed into the QMFD to localize tampered regions $M_{loc}$.

four blocks based on its structural design. The overall network utilizes both the RGB modality $X_{img} \in \mathbb{R}^{3 \times H \times W}$ and the high-frequency noise modality $X_{noise} \in \mathbb{R}^{3 \times H \times W}$ processed by the MNFM as input. The MNFM comprises SRM (Zhou et al., 2018), BayarConv (Bayar & Stamm, 2018), and NoisePrint++ (Guillaro et al., 2023), along with a fusion convolution, to aggregate features including local statistical anomalies, pixel-wise anomaly correlations, and device source noise.

Subsequently, the details of the Multi-stage Alternating Feature Extraction and Interaction are elaborated. At each stage, the dual-modal features $(R_i, N_i, i = 0, 1, 2, 3)$ from the previous level (where the initial input is regarded as level 0) are fed into the feature extraction blocks of the corresponding backbone networks, producing initial features for the next level $(R_{i+1}^{init}, N_{i+1}^{init})$:

$$R_{i+1}^{init} = GAE_{i+1}(R_i) \quad N_{i+1}^{init} = LAE_{i+1}(N_i) \tag{1}$$

The $R_{i+1}^{init}$ are aligned with the noise modality. Both sets of features are then fed into the DFCM for deep interaction (as illustrated in Fig 2):

$$(R_{i+1}, N_{i+1}) = DFCM_{i+1}(\delta(R_{i+1}^{init}), N_{i+1}^{init}) \tag{2}$$

Where $\delta$ represents cross-modal Feature Alignment. The overall structure of the DFCM is symmetrically designed. First, the dual-modal features undergo lightweight channel attention for feature recalibration $\Gamma$. Channel-wise aggregation of the features is performed via adaptive average pooling, followed by the acquisition of channel weights through a projection head. The input features are then multiplied by these channel weights to obtain the recalibrated feature representation $(R_{i+1}^{mid}, N_{i+1}^{mid})$:

$$R_{i+1}^{mid} = \Gamma_{i+1}^R(\delta(R_{i+1}^{init}) \quad N_{i+1}^{mid} = \Gamma_{i+1}^N(N_{i+1}^{init}) \tag{3}$$

Subsequently, cross-attention is performed between the two modal features. Each modality serves as the query $Q$ while the other provides the key-value pair $(K, V)$, enabling bidirectional retrieval of intrinsic relationships and valuable information from the modalities. This mechanism enhances both the complementary of the dual-modal features and their sensitivity to tampering artifacts. In contrast to classical cross-attention approaches, Group Convolution Modules (GCM) (Krizhevsky et al., 2012) are utilized as the transformation matrices for $Q$, $K$, and $V$, instead of applying simple

linear transformations to the features prior to performing attention. This design not only reduces parameter and computational costs but also enables the capture of diverse feature patterns, similar to multi-head attention. It provides a unified representation space for features originating from different modalities, allowing cross-attention to compute inter-feature correlations more effectively. Finally, the output is generated by combining the result of the cross-attention with a residual connection of the original features, ensuring that the original information remains preserved:

$$R_{i+1} = R_{i+1}^{init} + Attn_{N \leftarrow R}(G(R_{i+1}^{mid}), G(N_{i+1}^{mid}), G(N_{i+1}^{mid}))$$
$$N_{i+1} = N_{i+1}^{init} + Attn_{R \leftarrow N}(G(N_{i+1}^{mid}), G(R_{i+1}^{mid}), G(R_{i+1}^{mid}))$$

(4)

After being aligned back to the original feature space, the RGB modality features, together with the noise modality features, are returned to the corresponding backbone for the extraction of features at the next level.

### 3.3 QUERY-DRIVEN MULTI-LEVEL FEATURE DECODING MECHANISM

Existing approaches typically integrate upsampling with convolution or MLP. However, this strategy fails to pre-screen features, indiscriminately introducing substantial irrelevant background information and noise into the decoder. Consequently, subtle tampering features are severely overwhelmed. Inspired by the mechanism of the SAM Mask Decoder (Kirillov et al., 2023), we propose a lightweight, Query-driven Multi-level Feature Decoding. Specifically, a set of learnable tamper-aware queries $Q_{taq} \in \mathbb{R}^{N \times dim}$ is initialized at the beginning of the framework. These queries are subsequently utilized to generate corresponding query embeddings for each level of features. At each stage, the queries, together with the enhanced dual-modal features, are fed into the MFAM to be updated and generate the corresponding query embeddings $Q_i, i = 1, 2, 3, 4$:

$$Q_i = MFAM(Q_i^{init}, R_i, N_i)$$

(5)

Where $Q_i^{init}$ represents the initial state of the i-th level queries. The MFAM consists of three transformer blocks (Vaswani et al., 2017) (as shown in Fig. 2). The first two perform cross-attention, enabling the queries to establish richer and more robust cross-modal representations from both modalities while filtering out irrelevant features. The third block employs multi-head self-attention to further refine and integrate the information it has learned, thereby producing a final representation that is more semantically coherent and contextually enriched. Subsequently, the query embeddings $Q_i$ are passed through an MLP layer to generate the queries $Q_{i+1}^{init}$ for the next level, repeating the aforementioned process. The query embeddings at each level accumulate aggregated information from all preceding stages. This process is formulated as follows:

$$Q_i = Trans_{R \leftarrow Q}(Q_i^{init}, R_i, N_i)$$
$$Q_i = Trans_{N \leftarrow Q}(Q_i, N_i, Q_i)$$
$$Q_i = Trans_{Q \leftarrow Q}(Q_i, Q_i, Q_i)$$
$$Q_{i+1}^{init} = MLP(Q_i)$$

(6)

Then, the dual-modal features and their corresponding query embeddings from all levels are grouped into four triplets $(Q_i, R_i, N_i)$ and fed into the QMFD. Its internal architecture is illustrated in the appendix. Within the decoder, the RGB modality features and noise modality features from the same level are fed into a Dilated Convolution Module (DCM) Wang et al. (2018), leveraging its large receptive field to achieve preliminary fusion. The fused features $\{F_1^{fin} \sim F_4^{fin}\}$ are then multiplied with the corresponding query embeddings via matrix multiplication. This step is designed to perform feature selection. Finally, the features from all levels are upsampled to a common resolution and fed into the prediction head to output the final localization mask $M_{loc} \in \mathbb{R}^{1 \times H \times W}$:

$$M_{loc} = PredictHead(ConCat(Q_i \otimes F_i^{fin}, i = 1, 2, 3, 4))$$

(7)

### 3.4 LOSS FUNCTION DESIGN

To enable the network to focus more on the edges of tampered areas, the RGB modality features output at each stage are fed into an Edge Supervision Module to predict the boundaries of tampered regions, thereby constructing an edge loss $Loss_{edg}$. The predicted mask output from the decoder is

used to construct the segmentation loss $Loss_{seg}$. Considering the extreme imbalance between tampered and authentic pixels, Dice Loss (Milletari et al., 2016) is employed to compute both $Loss_{edg}$ and $Loss_{seg}$. The total loss $Loss_{total}$ of the model is formulated as follows, where, based on empirical practice, $\alpha$ is set to 0.2 and $\beta$ to 0.8.

$$Loss_{total} = \alpha \dot{L}oss_{seg} + \beta \dot{L}oss_{edg} \tag{8}$$

## 4 EXPERIMENT

### 4.1 EXPERIMENT SETUP

**Training Dataset and Implementation Details**  Our model is trained on the standardized Protocol-CAT dataset (Kwon et al., 2021), which consists of CASIAv2 (Dong et al., 2013), IMD2020 (Novozamsky et al., 2020), FantasticReality (Kniaz et al., 2019), and TampCOCO Kwon et al. (2022), totaling 825,997 images. These images cover multiple tampering types, such as splicing, copy-move, and removal. Each image is resized to $512 \times 512$ for training input. The model is trained for 150 epochs with a batch size of 32. We adopt a cosine decay learning rate schedule, initialized at 1e-4 and gradually reduced to a minimum of 5e-7. The AdamW optimizer is employed with a weight decay of 0.05 to mitigate overfitting. All models are trained and evaluated on IMDLBenco (Ma et al., 2024).

**Test Dataset and Evaluation Metric**  The evaluation of our model is conducted on a series of public benchmarks, encompassing four widely adopted datasets: CASIAv1 (Dong et al., 2013), Coverage (Wen et al., 2016), NIST16 (Guan et al., 2019), Columbia (Hsu & Chang, 2006). These collections comprise images that exhibit a wide range of resolutions and incorporate diverse tampering strategies. To quantitatively assess the model's performance in IML, we employ the pixel-level F1 and AUC score as the primary evaluation metrics.

Table 1: Comparison of Pixel-level F1 and AUC across four datasets. Best results are **bold**, second-best are underlined.

| Method | Pixel-level F1 | | | | | Pixel-level AUC | | | | |
|---|---|---|---|---|---|---|---|---|---|---|
| | **CAS** | **COl** | **COV** | **NIST** | **AVG** | **CAS** | **COl** | **COV** | **NIST** | **AVG** |
| ManTra-Net | 0.327 | 0.462 | 0.196 | 0.193 | 0.295 | 0.643 | 0.724 | 0.566 | 0.709 | 0.661 |
| PSCC-Net | 0.578 | 0.822 | 0.341 | 0.416 | 0.539 | 0.918 | 0.919 | 0.872 | 0.810 | 0.880 |
| MVSS-Net | 0.583 | 0.723 | 0.470 | 0.320 | 0.524 | 0.904 | 0.911 | 0.868 | 0.777 | 0.865 |
| CAT-Net | 0.778 | 0.923 | 0.485 | 0.450 | 0.659 | 0.965 | 0.962 | 0.907 | 0.867 | 0.925 |
| TruFor | 0.700 | 0.903 | 0.379 | 0.426 | 0.602 | 0.951 | 0.936 | 0.887 | 0.863 | 0.909 |
| IML-ViT | 0.751 | 0.927 | 0.546 | 0.140 | 0.591 | 0.961 | 0.941 | 0.921 | 0.812 | 0.909 |
| SAM | 0.627 | 0.817 | 0.401 | **0.509** | 0.589 | 0.945 | **0.973** | 0.886 | 0.876 | 0.920 |
| Mesorch | 0.826 | 0.905 | 0.526 | 0.412 | 0.667 | 0.979 | 0.924 | 0.917 | **0.891** | 0.928 |
| QMA-Net(Ours) | **0.873** | **0.939** | **0.659** | 0.480 | **0.738** | **0.985** | 0.943 | **0.931** | 0.860 | **0.930** |

### 4.2 PERFORMANCE COMPARISON WITH STATE-OF-THE-ART

We adopt ManTra-Net (Wu et al., 2019), PSCC-Net (Liu et al., 2022), CAT-NetKwon et al. (2022), MVSS-Net (Dong et al., 2023), TruFor (Guillaro et al., 2023), IML-ViT (Ma et al., 2023), SAM, and Mesorch (Zhu et al., 2025) as baseline methods for comparison. For fairness, all baseline models are retrained on the Protocol-CAT dataset. The corresponding experimental results are reported in Table 1. As shown in the table, QMA-Net consistently surpasses the existing state-of-the-art methods in IML across the four benchmark datasets. In addition, Fig. 3 illustrates a qualitative comparison of the predicted results from our model and the competing approaches. It can be clearly observed that our framework delineates the boundaries of manipulated regions more accurately, leading to fewer false alarms and higher precision. This fact demonstrates that our framework successfully constructs complementary dual-modal representations sensitive to tampering and accurately localizes the tampered regions.

Figure 3: IML results on multiple datasets. The leftmost two columns are the manipulated image and groundtruth, followed by the prediction results of different models.

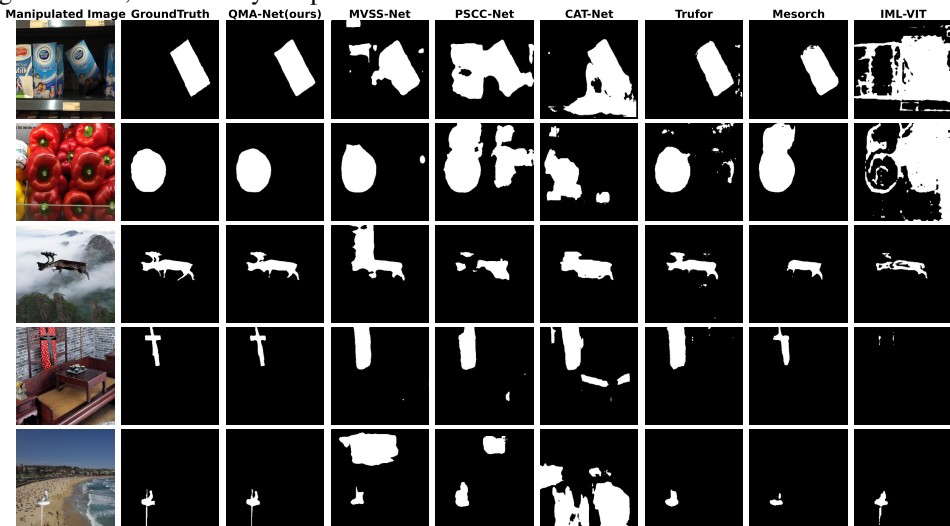

Figure 4: Robustness test results. The x-axis represents the attack intensity, while the y-axis denotes the pixel-level F1 score on the corresponding test datasets.

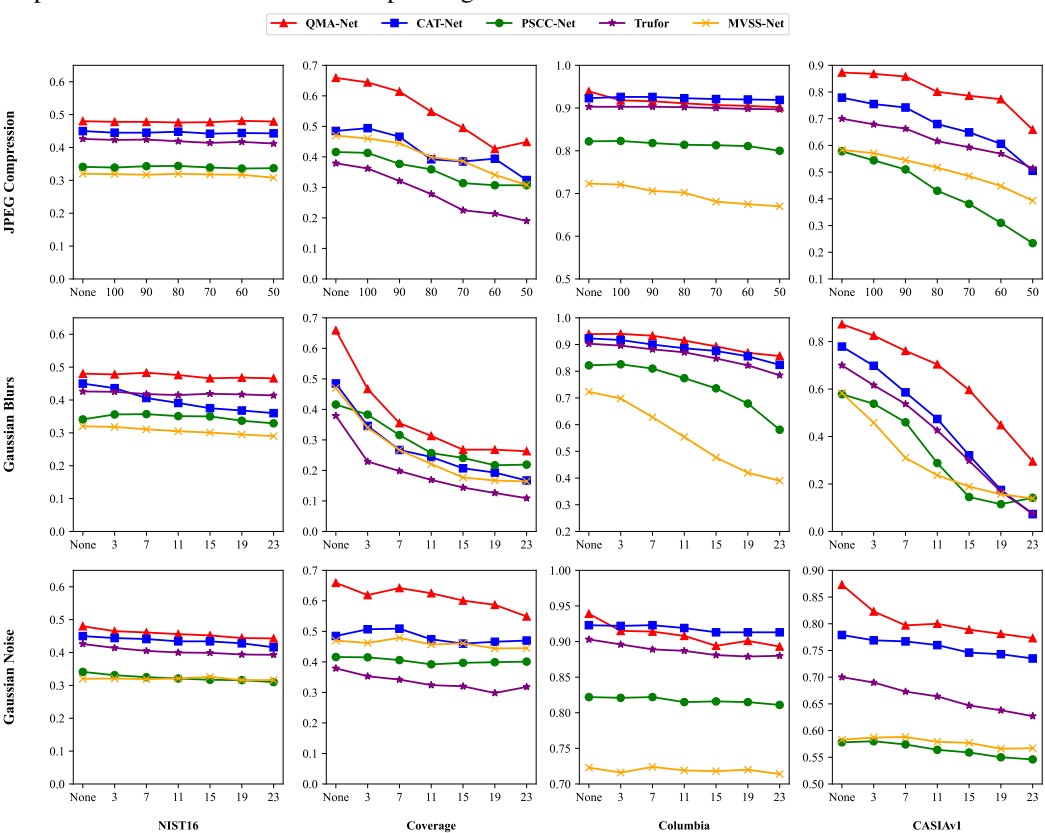

## 4.3 ROBUSTNESS STUDY

To assess the robustness of the model against different attack scenarios, we apply degradation operations—including Gaussian noise, Gaussian blur, and JPEG compression—to the tampered images. The corresponding results are depicted in Fig. 4. Our framework consistently surpasses other SoTA approaches on the CASIA, Coverage, and NIST16 datasets across all attack types. For the Columbia, our method achieves superior performance under Gaussian blur and is only marginally outperformed by CAT-Net under the other two degradations. These findings collectively demonstrate the strong robustness of our model.

## 4.4 ABLATION STUDY

Table 2: Ablation Study Results. We conduct five additional sets of experiments to validate the effectiveness of the proposed components.

| Method | Setting | Pixel-level F1 Score | | | | |
|---|---|---|---|---|---|---|
| | | CAS | COL | COV | NIST | AVG |
| Ours w/ CP | w/o DFCM | 0.850 | 0.943 | 0.579 | 0.456 | 0.707 |
| Ours w/ SL | Only $(Q_4, R_4, N_4)$ | 0.849 | 0.943 | 0.604 | 0.416 | 0.703 |
| Ours w/ SS | Only Stage#4 | 0.850 | 0.919 | 0.592 | 0.444 | 0.701 |
| Ours w/ CD | Conv Decoder | 0.857 | 0.946 | 0.582 | 0.438 | 0.706 |
| Ours w/ MLP | MLP Decoder | 0.859 | 0.936 | 0.601 | 0.445 | 0.710 |
| QMA-Net | Ours | **0.873** | **0.939** | **0.659** | **0.480** | **0.738** |

**Ablation study on Multi-stage Alternating Feature Extraction and Interaction** We validate the effectiveness of the proposed Multi-stage Alternating Feature Extraction and Interaction paradigm from two perspectives. On one hand, we remove DFCM at each stage; on the other hand, we only retain the fourth stage. The results are listed in Settings 1 and 3 of Table 2. respectively. Our model demonstrates an average performance improvement of 4.2% and 4.7% compared to these two scenarios, respectively. This fact indicates that our method effectively constructs complementary bimodal tampering representations, enhances sensitivity to tampering artifacts, and suppresses intra-modal noise.

**Ablation study on Query-driven Multi-level Feature Decoding** We replace the Query-driven Multi-level Feature Decoding with upsampling followed by convolutional or MLP decoders. The results are shown in Settings 4 and 5 in Table 2. Our model shows improvements of 4.5% and 3.9% compared to these two scenarios, respectively. Furthermore, compared to using single-level features (setting 2), our model achieves a 5.0% performance improvement. These findings indicate the necessity of each level of features and that our method achieves effective feature selection and aggregation.

## 5 CONCLUSION

In this work, we propose a novel IML network, QMA-Net, which consists of a Multi-stage Alternating Feature Extraction and Interaction architecture and a lightweight, Query-driven Multi-level Feature Decoding. The former simulates the cognitive processes of the human brain, constructing complementary tampering feature representations through cross-modal Feature Alignment and DFCM at each stage, thereby enhancing sensitivity to tampering artifacts. Moreover, the latter employs learnable tamper-aware queries to progressively aggregate crucial information from multi-level features through MFAM at each stage. In QMFD, these query embeddings perform selective refinement and aggregation of multi-level features to accurately predict tampered regions. Extensive experiments demonstrate that our framework outperforms current SoTA models in localization accuracy and robustness across multiple public datasets. Simultaneously, our model exhibits a reduction in both parameter count and FLOPs.

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

# A  APPENDIX

## A.1  DETAILED INFORMATION ABOUT THE DATASETS USED IN THIS PAPER

We trained our model using the protocol-CAT dataset, which was first introduced and utilized by CAT-Net. It consists of four datasets: CASIAv2, IMD2020, TampCOCO and Fantastic Reality. CASIA2.0 provides high-quality copy-move and spliced tampered images. IMD2020 includes complex real-world edits (such as splicing and local modifications) with non-fixed resolutions. Fantastic Reality is a multi-task annotation dataset that integrates tamper localization and semantic segmentation, providing pixel-level tamper region masks, instance segmentation, and category labels. TampCOCO is constructed based on the COCO 2017 dataset and includes two parts: copy-move and splicing. All images undergo JPEG compression while retaining clear boundaries to support model learning of low-level tampering traces. The information of all training sets is listed in Table 3. These training sets contain rich semantic details and noise patterns, making them more aligned with real-world application scenarios. By training on these datasets, we can learn more comprehensive and hierarchical features of tampering, effectively enhancing both robustness and generalization capabilities.

We select six public benchmark datasets as our test data, namely CAISAv1, Columbia, Coverage, NIST16, COCOGlide, and AutoSplice. CASIAv1 primarily provides high-quality spliced image. The Columbia dataset focuses on uncompressed spliced image and features high-resolution. The Coverage dataset addresses copy-move forgeries, typically by copying and pasting one item from a group of similar objects within an image.The NIST16 comprises three forgery types: splicing, removal, and copy-move operations, and maintains high image resolution throughout. The COCOGlide focuses on generative image forgery research by combining the GLIDE diffusion model with semantic prompts to create tampered content, simulating semantic-level local manipulations. The AutoSplice represents a text-prompt manipulated image collection where all images undergo JPEG compression processing. The information of all test sets is listed in Table 4. These datasets encompass diverse manipulation types, exhibit wide resolution ranges, and contain varied forgery region sizes, collectively enabling comprehensive evaluation of model performance across multiple dimensions.

Table 3: Detailed information of the protocol-CAT training set, where "N/K" indicates that the quantity of this type is unknown.

| Dataset | Manipulation type | | | Number | Resolution | |
|---|---|---|---|---|---|---|
| | cope-move | splice | remove | | min | max |
| CASIAv2 | 3274 | 1849 | 0 | 5123 | $320 \times 240$ | $800 \times 600$ |
| IMD2020 | N/K | N/K | N/K | 2010 | $260 \times 193$ | $2958 \times 4437$ |
| TampCOCO | 600000 | 200000 | 0 | 800000 | $72 \times 51$ | $640 \times 640$ |
| Fantastic Reality | N/K | N/K | N/K | 19423 | $500 \times 333$ | $6000 \times 4000$ |
| Total | N/K | N/K | N/K | 826556 | $72 \times 51$ | $6000 \times 4000$ |

## A.2  THE DETAILS OF QMFD

The internal architecture of QMFD is illustrated in Fig. 5. The QMFD comprises four branches, each dedicated to processing dual-modal features at four distinct levels. Given a ternary tuple $(Q_i, R_i, N_i)$ at one level, the dual-modal features are fused through the DCM to produce the fused features

Table 4: Detailed information of the six public benchmark datasets, where "N/K" indicates that the quantity of this type is unknown.

| Dataset | Manipulation type | | | | Number | Resolution | |
| --- | --- | --- | --- | --- | --- | --- | --- |
| | cope-move | splice | remove | AI-Gen | | min | max |
| CASIAv1 | 459 | 461 | 0 | 0 | 920 | $384 \times 256$ | $384 \times 256$ |
| Columbia | 0 | 180 | 0 | 0 | 180 | $757 \times 568$ | $1152 \times 768$ |
| Coverage | 100 | 0 | 0 | 0 | 100 | $334 \times 190$ | $752 \times 472$ |
| NIST16 | 236 | 225 | 103 | 0 | 564 | $500 \times 500$ | $5616 \times 3744$ |
| COCOGlide | 0 | 0 | 0 | 512 | 512 | $256 \times 256$ | $256 \times 256$ |
| AutoSplice | 0 | 0 | 0 | 3621 | 3621 | $256 \times 256$ | $4232 \times 4232$ |
| Total | 795 | 866 | 103 | 4133 | 5897 | $334 \times 190$ | $5616 \times 3744$ |

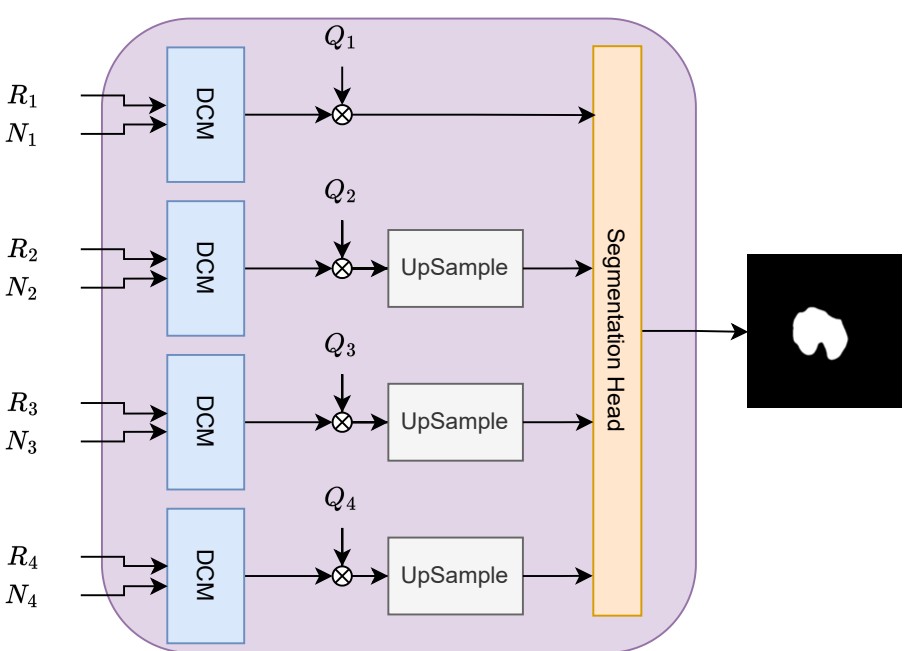

Figure 5: The pipeline of QMFD. A DCM and multi-level query embeddings are leveraged to pre-screen features, followed by concatenation of features from all levels into the prediction head for precise localization of tampered regions.

$F_i^{fin}$. The query $Q_i$ then performs pixel-wise feature weighting and reconstruction across the $F_i^{fin}$. Specifically, each query shares the same length as the pixel's feature vector, enabling element-wise weighted summation operations. This step aims to emphasize critical information while suppressing irrelevant feature interference. Finally, the refined features from all four levels are fed into a prediction head to output the tampered region mask. The prediction head consists of a simple $1 \times 1$ convolutional layer.

### A.3 COMPARSION WITH BASELINE MODELS USING ADDITIONAL METRIC

We conduct supplementary experiments on four benchmark datasets using permute-F1 and pixel-level IoU scores as evaluation metrics. The results are presented in Table 5. As shown in the results, QMA-Net achieves SoTA performance across all metrics, with particularly significant improvements in average permute-F1 and pixel-level IoU scores compared to other models. This outcome further

validates that our method successfully constructs complementary dual-modal features sensitive to tampering artifacts, and effectively filters and aggregates multi-level features through the query-driven mechanism.

Table 5: Comparison of permute-F1 and Pixel-level IoU across four datasets. Best results are **bold**, second-best are underlined.

| Method | Permute-F1 | | | | | Pixel-level IoU | | | | |
|---|---|---|---|---|---|---|---|---|---|---|
| | CAS | COl | COV | NIST | AVG | CAS | COl | COV | NIST | AVG |
| PSCC-Net | 0.559 | 0.830 | 0.451 | 0.371 | 0.553 | 0.442 | 0.729 | 0.307 | 0.259 | 0.434 |
| MVSS-Net | 0.597 | 0.768 | 0.529 | 0.357 | 0.563 | 0.481 | 0.641 | 0.397 | 0.236 | 0.439 |
| CAT-Net | 0.786 | 0.951 | 0.551 | 0.490 | 0.695 | 0.716 | 0.903 | 0.430 | 0.388 | 0.609 |
| TruFor | 0.714 | 0.934 | 0.443 | 0.466 | 0.639 | 0.621 | 0.874 | 0.303 | 0.350 | 0.537 |
| IML-ViT | 0.763 | 0.974 | 0.595 | 0.169 | 0.625 | 0.683 | 0.919 | 0.482 | 0.127 | 0.553 |
| Mesorch | 0.837 | 0.966 | 0.594 | 0.467 | 0.716 | 0.778 | 0.896 | 0.481 | 0.353 | 0.627 |
| QMA-Net | **0.880** | **0.995** | **0.697** | **0.520** | **0.773** | **0.832** | 0.935 | **0.617** | **0.420** | **0.701** |

## A.4 THE MODEL'S PERFORMANCE ON AI-GENERATED TAMPERING TECHNIQUES

We supplement the performance of QMA-Net on CocoGlide and AutoSplice (image tampering based on diffusion models or LLM) to evaluate localization capabilities for AI-based tampered image. As shown in the Table 6, QMA-Net achieves optimal or sub-optimal performance across all metrics on both datasets.

Table 6: Comparison on COCOGlide and AutoSplice under with different SoTA models. Best results are **bold**, second-best are underlined.

| Method | COCOGlide | | | AutoSplice | | |
|---|---|---|---|---|---|---|
| | F1 | AUC | IoU | F1 | AUC | IoU |
| MVSS-Net | 0.428 | 0.819 | 0.327 | 0.388 | 0.755 | 0.272 |
| PSCC-Net | 0.458 | 0.848 | 0.396 | **0.455** | 0.871 | **0.406** |
| CAT-Net | 0.409 | 0.849 | 0.334 | 0.450 | 0.862 | 0.348 |
| IML-ViT | 0.369 | 0.835 | 0.290 | 0.343 | 0.854 | 0.246 |
| Mesorch | 0.397 | **0.894** | 0.329 | 0.357 | **0.926** | 0.252 |
| QMA-Net | **0.477** | 0.867 | **0.416** | 0.451 | 0.875 | 0.352 |

## A.5 QUANTITATIVE RESULTS OF ROBUSTNESS STUDY

We quantify the robustness test data presented in Figure 4, using the average pixel-level F1 score under single-attack types with varying intensity factors as the evaluation metric. The results are presented in Tables 6 and 7. We apply degradation techniques such as Gaussian noise (GN) with different standard deviations(3,7,11,15,19,23), Gaussian blur (GB) with varying kernel sizes (3,7,11,15,19,23), and JPEG compression (JC) with different quality factors (100,90,80,70,60,50) to the tampered images. As visually demonstrated in both tables, QMA-Net achieves SoTA performance on CASIAv1, Coverage, and NIST16 datasets across all attack types.

## A.6 GRAD-CAM ANALYSIS OF MULTI-LEVEL FEATURES

We visualized the fused multi-level features using Grad-CAM in Figure. 6. Red areas represent the high-response regions in the feature maps. It can be observed that Level 1 and Level 2 features focus on the edges of the tampered regions, effectively capturing low-level features. In contrast, the deeper Level 3 and Level 4 features extract high-level object-based characteristics without interference from other regions. This phenomenon demonstrates that our model successfully constructs complementary dual-modal features sensitive to tampering artifacts, and effectively aggregates these

Table 7: Avergae pixel-level F1 comparison on CASIAv1 and Columbia datasets under different attacks. Best results are **bold**, second-best are underlined.

| Method | CASIAv1 | | | Columbia | | |
|---|---|---|---|---|---|---|
| | JC | GB | GN | JC | GB | GN |
| PSCC-Net | 0.427 | 0.324 | 0.564 | 0.814 | 0.747 | 0.817 |
| MVSS-Net | 0.506 | 0.296 | 0.578 | 0.697 | 0.556 | 0.719 |
| CAT-Net | 0.674 | 0.444 | 0.757 | **0.923** | 0.883 | **0.918** |
| TruFor | 0.619 | 0.403 | 0.663 | 0.901 | 0.858 | 0.888 |
| QMA-Net | **0.803** | **0.643** | **0.805** | 0.914 | **0.907** | 0.909 |

Table 8: Average pixel-level F1 comparison on NIST16 and Coverage datasets under different attacks. Best results are **bold**, second-best are underlined.

| Method | NIST16 | | | Coverage | | |
|---|---|---|---|---|---|---|
| | JC | GB | GN | JC | GB | GN |
| PSCC-Net | 0.340 | 0.346 | 0.323 | 0.356 | 0.293 | 0.404 |
| MVSS-Net | 0.317 | 0.306 | 0.320 | 0.401 | 0.258 | 0.460 |
| CAT-Net | 0.445 | 0.398 | 0.435 | 0.420 | 0.273 | 0.482 |
| TruFor | 0.419 | 0.419 | 0.404 | 0.281 | 0.193 | 0.333 |
| QMA-Net | **0.478** | **0.474** | **0.457** | **0.548** | **0.370** | **0.612** |

dual-modal features through query-driven mechanisms, thereby suppressing interference from irrelevant features.

### A.7 ABLATION STUDY ON QUERY QUANTITY

We vary the number of queries in the framework by adjusting them to three configurations: 4, 8, and 32, to investigate the impact of query quantity on model performance. The experimental results are shown in Table 9. Our model (16 queries) demonstrates improvements of 2.8%, 2.4%, and 3.1% in average p-F1 scores compared to the configurations with 4, 8, and 32 queries respectively. It is noted that the model's performance does not monotonically improve with increasing query quantity, but rather follows a unimodal curve pattern.

Table 9: The results of ablation study on query quantity. Best results are **bold**, second-best are underlined.

| Queries | Pixel-level F1 Score | | | | |
|---|---|---|---|---|---|
| | CAS | COl | COV | NIST | AVG |
| 4 queries | 0.864 | 0.942 | 0.613 | 0.450 | 0.717 |
| 8 queries | 0.857 | 0.933 | 0.641 | 0.448 | 0.720 |
| 32 queries | 0.861 | **0.952** | 0.590 | 0.457 | 0.715 |
| 16 queries(Ours) | **0.873** | 0.939 | **0.659** | **0.480** | **0.738** |

### A.8 FLOPS AND PARAMETERS

The number of parameters and FLOPs for all measurements was calculated based on a batch size of 1. As shown in Table 10, our model has a comparable computational burden to VLMs-based models while demonstrating higher accuracy. Our model achieves SoTA performance while requiring significantly fewer parameters and lower computational overhead (FLOPs). Notably, the FLOPs of our model are nearly seven times lower than those of vision foundation model-based approaches (e.g., IMDprompt). These results demonstrate the lightweight nature and superior practical value of our proposed method.

Table 10: Comparison of parameters and computational efficiency (Flops) across different models.

| Method | Parameters (M) | FLOPs (G) |
|---|---|---|
| ManTra-Net | 3.9 | 274.0 |
| MVSS-Net | 150.5 | 171.0 |
| PSCC-Net | 3.7 | 376.8 |
| CAT-Net | 116.7 | 137.2 |
| TruFor | 68.7 | 236.5 |
| SAM | 309.0 | 1499.0 |
| IMDPrompt | 347.6 | 1533.0 |
| QMA-Net (Ours) | 114.0 | 230.0 |

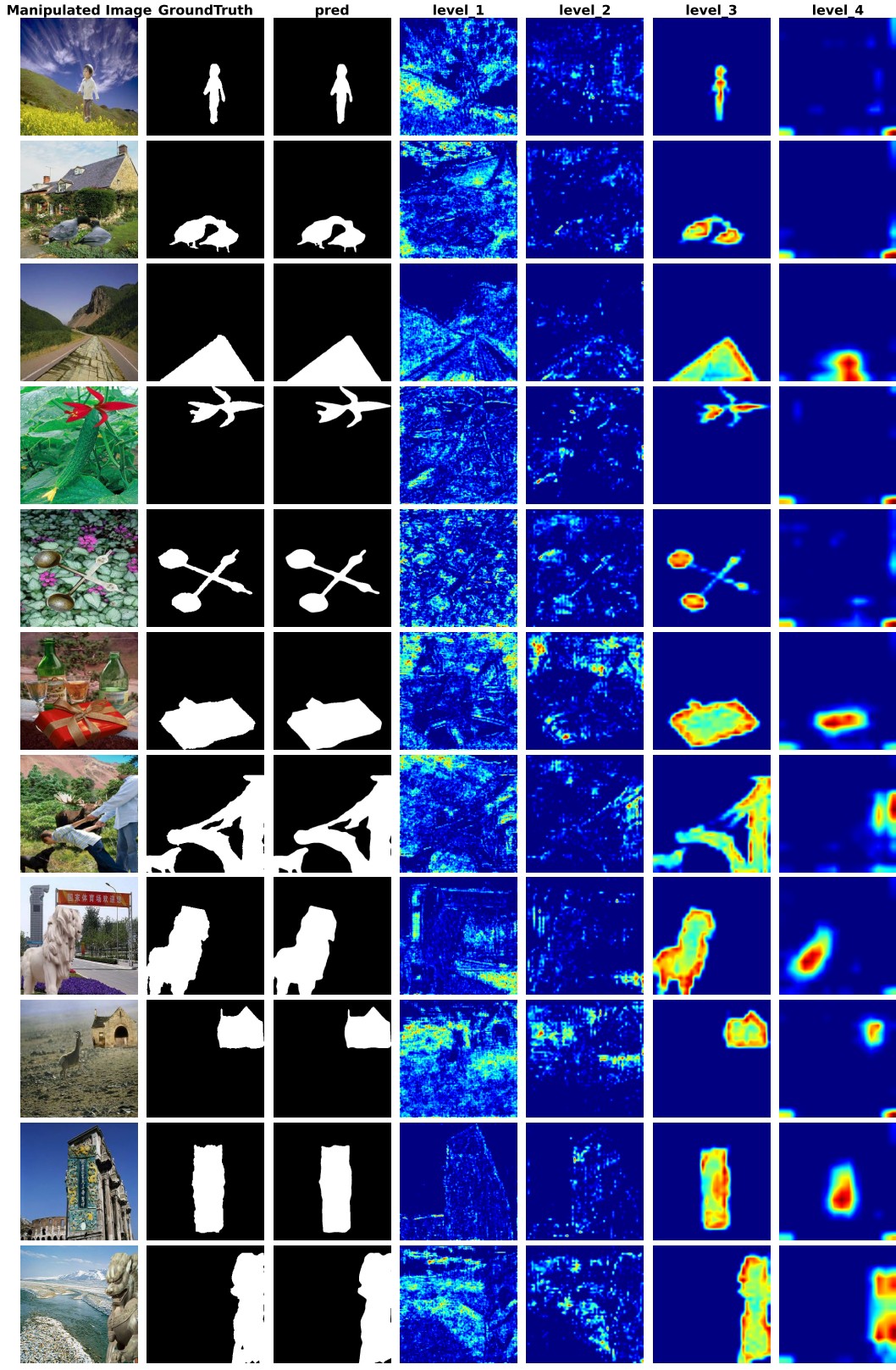

Figure 6: Visualization of Grad-CAM for multi-level features. The first three columns are the tampered image, groundtruth, and prediction results, respectively. Followed by the corresponding fused multi-level features.

