# OpenReview forum: "A Novel Query-Driven Multi-Stage Alternating Feature Extraction and Interaction Network for Image Manipulation Localization"
_ICLR.cc/2026/Conference — Submitted to ICLR 2026_

### Official Review · Reviewer_AiRe · 2025-10-15

**Soundness:** 2
**Presentation:** 2
**Contribution:** 1
**Rating:** 2
**Confidence:** 5

**Summary:**

This paper addresses the insufficient modal interaction in existing dual-stream networks for Image Manipulation Localization (IML) by proposing a novel framework named QMA-Net. The core of this framework is a Multi-stage Alternating Feature Extraction and Interaction architecture, which deeply fuses RGB and noise modality information throughout multiple stages rather than performing a single fusion at the end. Building upon this, the paper further designs a lightweight, Query-driven Multi-level Feature Decoder that employs learnable "tamper-aware queries" to actively screen and aggregate critical tampering cues from multi-level features. Experimental results demonstrate that this method achieves competitive performance on several public IML benchmarks.

**Strengths:**

- The paper proposes a structured, multi-stage fusion framework that attempts to address the common issue of insufficient modal interaction in dual-stream networks. The authors validate their method through systematic experiments, evaluating it on multiple public datasets and using detailed ablation studies to analyze the contributions of its key components.
- The experimental results show that the proposed method outperforms several existing baseline models across multiple evaluation metrics. This indicates that the proposed components, when combined, form an effective manipulation localization system, demonstrating its potential for this specific task.

**Weaknesses:**

- The paper successfully demonstrates the effectiveness of the "tamper-aware queries" but fails to provide a deep exploration of their working mechanism. It is unclear what patterns these query vectors learn or whether they exhibit preferences for different artifacts (e.g., edges, textures, semantics) at different network levels. The lack of such visualization or quantitative analysis significantly undermines the model's interpretability.
- QMA-Net introduces several elaborate modules (e.g., DFCM), which increase the model's overall complexity. Although ablation studies show that removing these modules degrades performance, the paper lacks a comparison with simpler alternatives (e.g., using simple feature concatenation or addition for interaction within the same multi-stage framework). This makes it difficult for readers to determine whether the current design's complexity is necessary to achieve high performance.
- The paper mentions that its noise extraction front-end (MNFM) fuses multiple features from SRM, BayarConv, and NoisePrint++. While this is a powerful combination, the paper does not detail how these features are fused, nor does it experimentally analyze their individual contributions. An internal ablation study of the MNFM module would make the methodology description more complete and rigorous.

**Questions:**

- The ablation study in the appendix (Table 9) shows that performance degrades when the number of queries is increased from 16 to 32. Could you provide an explanation for this interesting non-monotonic phenomenon? Does this suggest that image manipulation features are inherently sparse, causing an excessive number of queries to introduce redundant information or noise that interferes with the final localization?
- Could you quantitatively explain the specific advantages of feeding the interacted features back into the backbone for the next stage of extraction? How much would performance drop if a simpler architecture were used—one without feedback that simply aggregates the fused features from all stages at the decoder end? This would help us better understand the core value of your design.

---

> ### Author Response · Authors · 2025-11-21
> **Response to Reviewer AiRe**
>
> # Response to Reviewer AiRe
>
> Thank you for your comments. We will respond to your questions point by point.
>
> Q1: QMA-Net introduces several elaborate modules (e.g., DFCM), which increase the model's overall complexity. Although ablation studies show that removing these modules degrades performance, the paper lacks a comparison with simpler alternatives (e.g., using simple feature concatenation or addition for interaction within the same multi-stage framework). This makes it difficult for readers to determine whether the current design's complexity is necessary to achieve high performance.
>
> A: We replaced the DFCM with concatenation and element-wise addition and retrained the model. The results are shown below (pixel-F1). It can be observed that, compared to these simpler methods, the DFCM does not introduce significant additional parameters or computational complexity, yet achieves the best performance. This advantage is attributed to our deliberate design.
>
> | Interaction method    | Parameters (M) | FLOPS (G) | CASIA1.0  | Columbia  | NIST16    | Coverage  | Avg.F1    |
> | --------------------- | -------------- | --------- | --------- | --------- | --------- | --------- | --------- |
> | DFCM                  | 114.0          | 230.0     | **0.873** | 0.939     | **0.480** | **0.659** | **0.738** |
> | Element-wise addition | **105.0**      | **227**   | 0.867     | 0.925     | 0.468     | 0.616     | 0.719     |
> | Concatenation         | 112.0          | 232.0     | 0.862     | **0.951** | 0.448     | 0.628     | 0.722     |
>
> Q2: The paper mentions that its noise extraction front-end (MNFM) fuses multiple features from SRM, BayarConv, and NoisePrint++. While this is a powerful combination, the paper does not detail how these features are fused, nor does it experimentally analyze their individual contributions. An internal ablation study of the MNFM module would make the methodology description more complete and rigorous.
>
> A: In the MNFM, we posit that a simple 3x3 convolution is sufficient to fuse the information derived from different noise extractors. We conducted three additional sets of experiments (using only SRM, Bayar, or NoisePrint++) to investigate the individual contribution of each feature extractor. The results are presented below (pixel-F1). They show that SRM contributes the most, followed by NoisePrint++, with Bayar having the least impact.
>
> | MNFM                  | CASIA1.0  | Columbia  | NIST16    | Coverage  | Avg.F1    |
> | --------------------- | --------- | --------- | --------- | --------- | --------- |
> | SRM+Bayar+NP++ (Ours) | 0.873     | 0.939     | **0.480** | **0.659** | **0.738** |
> | SRM                   | **0.874** | **0.960** | 0.449     | 0.620     | 0.726     |
> | Bayar                 | 0.862     | 0.944     | 0.443     | 0.580     | 0.707     |
> | NP++                  | 0.844     | 0.922     | 0.446     | 0.633     | 0.711     |
>
> Q3: The ablation study in the appendix (Table 9) shows that performance degrades when the number of queries is increased from 16 to 32. Could you provide an explanation for this interesting non-monotonic phenomenon? Does this suggest that image manipulation features are inherently sparse, causing an excessive number of queries to introduce redundant information or noise that interferes with the final localization?
>
> A: We also observed this phenomenon during our experiments. The non-semantic features of an image, such as frequency and noise, are highly sensitive to manipulation and show greater independence across different regions of the image[1]. We hypothesize that the underlying reason is that an excessive number of queries hinders the model's ability to effectively capture these non-semantic features. The limited tampering feature signals are forced to disperse across a larger number of query vectors. As a result, each query may only capture subtler, more fragmented cues, failing to form a comprehensive and highly discriminative representation of the tampered regions.

---

> > ### Author Response · Authors · 2025-11-30
> > **Response to Reviewer AiRe**
> >
> > # Response to Reviewer AiRe
> > Regarding your response, I must clarify the following points:
> >
> > - In our rebuttal to Reviewer 4Eo6 and manuscript itself, we have clearly explained the fundamental differences between our method's core mechanism—the Multi-stage Alternating Feature Extraction and Interaction architecture, along with the Query-driven Multi-level Feature Decoding—and the classic two-branch fusion frameworks, such as MVSS and Mesoch.
> > - To investigate the behavior of our multi-level queries, we employed Grad-CAM visualizations in Appendix 6 to illustrate the image regions attended to by each query level. It can be observed that low-level queries focus on features such as noise inconsistency and the edges of tampered regions, while high-level queries concentrate on the semantic features within the tampered area. This evidence effectively elucidates the behavior of the multi-level query mechanism, thereby enhancing the interpretability of our proposed method.

---

> ### Author Response · Authors · 2025-11-21
> **Response to Reviewer AiRe**
>
> # Response to Reviewer AiRe
> Q4: Could you quantitatively explain the specific advantages of feeding the interacted features back into the backbone for the next stage of extraction? How much would performance drop if a simpler architecture were used—one without feedback that simply aggregates the fused features from all stages at the decoder end? This would help us better understand the core value of your design.
>
> A: We have already conducted an ablation study for this scenario in Setting 1 of Table 2. When the DFCM is removed, features from each stage are fed directly into the decoder, where they are aggregated by the tamper-aware queries, and then the tamper mask is output. A comparison of the results is presented below. It can be observed that the method, which "simply aggregates the fused features from all stages at the decoder end without feedback" suffers a performance degradation of 4.4% compared to our approach. Meanwhile, Figure 1 also demonstrates the difference in sensitivity to tampering artifacts between the tampering features constructed under the two scenarios.
>
> | Method         | CASIA1.0  | Columbia  | NIST16    | Coverage  | Avg.F1    |
> | -------------- | --------- | --------- | --------- | --------- | --------- |
> | w/ DFCM (Ours) | **0.873** | **0.939** | **0.480** | **0.659** | **0.738** |
> | w/o DFCM       | 0.850     | 0.943     | 0.456     | 0.579     | 0.707     |
>
> We sincerely appreciate your comments and hope that our explanations and extra experiment results can effectively address your concerns. We would be grateful if you could consider raising the scores.
>
> [1] Su, Lei, et al. "Can we get rid of handcrafted feature extractors? sparsevit: Nonsemantics-centered, parameter-efficient image manipulation localization through spare-coding transformer." *Proceedings of the AAAI Conference on Artificial Intelligence*. Vol. 39. No. 7. 2025.

---

### Official Review · Reviewer_XaPB · 2025-10-31

**Soundness:** 3
**Presentation:** 2
**Contribution:** 2
**Rating:** 2
**Confidence:** 4

**Summary:**

This paper proposes QMA-Net, a query-driven multi-stage framework for Image Manipulation Localization (IML). It addresses two core limitations of existing methods: insufficient interaction between RGB (semantic boundaries) and high-frequency noise (tampering artifacts like compression errors) modalities, and redundant multi-level features. QMA-Net consists of two key components: (1) a Multi-stage Alternating Feature Extraction and Interaction module, which alternates "feature extraction–deep interaction" via cross-modal alignment and a Dual-modal Feature Cross-guided Module (DFCM) to build complementary bimodal representations; (2) a lightweight Query-driven Multi-level Feature Decoding module, which uses learnable tamper-aware queries to filter irrelevant information and aggregate critical features via a Multi-domain Feature Aggregation Module (MFAM). Experiments on 4 public datasets show QMA-Net outperforming baselines like CAT-Net and TruFor.

**Strengths:**

1. Unlike traditional "extract-then-simple-fusion" methods, QMA-Net enables stage-wise alternating interaction between RGB and noise modalities.
2. Learnable tamper-aware queries (16 queries, verified optimal via experiments) dynamically aggregate multi-level key information via MFAM, filtering background noise.
3. Experiments cover traditional tampering (splicing, copy-move) and AI-generated tampering (COCOGlide, AutoSplice), with robustness tests under Gaussian blur/JPEG compression. Results consistently outperform baselines.

**Weaknesses:**

1. Comparisons lack state-of-the-art methods (e.g., IMDPrompter, FakeShield) that excel in fine-grained or AI-generated tampering. This fails to demonstrate QMA-Net’s breakthrough.
2. Insufficient Theoretical Support for Key Designs: Critical choices (e.g., "RGB-aligned-to-noise" interaction order, DFCM’s Group Convolution grouping number) rely on empirical settings without quantitative analysis (e.g., gradient variance tests) or alternative design comparisons.
3. It is hard to clearly understand the key mechanisms from Fig.2.

**Questions:**

1. Comparisons lack state-of-the-art methods (e.g., IMDPrompter, FakeShield) that excel in fine-grained or AI-generated tampering. This fails to demonstrate QMA-Net’s breakthrough.
2. Insufficient Theoretical Support for Key Designs: Critical choices (e.g., "RGB-aligned-to-noise" interaction order, DFCM’s Group Convolution grouping number) rely on empirical settings without quantitative analysis (e.g., gradient variance tests) or alternative design comparisons.
3. It is hard to clearly understand the key mechanisms from Fig.2.

---

> ### Author Response · Authors · 2025-11-21
> **Response to Reviewer XaPB**
>
> # Response to Reviewer XaPB
>
> Thank you for your comments. We will respond to your questions point by point.
>
> Q1: Comparisons lack state-of-the-art methods (e.g., IMDPrompter, FakeShield) that excel in fine-grained or AI-generated tampering. This fails to demonstrate QMA-Net’s breakthrough.
>
> A: We retrained APSC-Net[1] and SparseViT[2] using the same training settings, and the test results are as follows (pixel-F1). The results for IMDPrompter[3] and FakeShield[4] are cited from the original paper.
>
> | Method         | CASIA1.0  | Columbia  | NIST16    | Coverage  | Avg.F1    |
> | -------------- | --------- | --------- | --------- | --------- | --------- |
> | APSC-Net       | 0.798     | 0.941     | **0.500** | 0.402     | 0.660     |
> | SparseViT      | 0.827     | **0.959** | 0.384     | 0.513     | 0.671     |
> | IMDPrompter    | 0.763     | 0.873     | 0.411     | 0.636     | 0.671     |
> | FakeShield     | 0.600     | 0.750     | 0.370     | \         | 0.573     |
> | QMA-Net (Ours) | **0.873** | 0.939     | 0.480     | **0.659** | **0.738** |
>
> It is worth noting that comparing our model with approaches like IMDPrompter and FakeShield is not entirely fair. This is because these models typically have extremely high parameter counts and FLOPs, resulting in very slow inference speeds. For instance, IMDPrompter has three times the parameters and nearly eight times the FLOPs of QMA-Net. FakeShield's metrics in these aspects are even significantly larger than those of IMDPrompter. While a higher parameter count should theoretically lead to better performance, our method still achieves SOTA performance, which sufficiently demonstrates its effectiveness and parameter efficiency.
>
> Q2: Insufficient Theoretical Support for Key Designs: Critical choices (e.g., "RGB-aligned-to-noise" interaction order, DFCM’s Group Convolution grouping number) rely on empirical settings without quantitative analysis (e.g., gradient variance tests) or alternative design comparisons.
>
> A: We retrained the model by changing the interaction order to "noise-aligned-to-RGB", and the results (pixel-F1) show that the "RGB-aligned-to-noise" interaction order yields better performance. We hypothesize that this is because noise features are compact, whereas RGB features contain a wealth of detailed information. Attempting to align noise features to match RGB features may introduce a significant amount of irrelevant or misleading information.
>
> | Interaction order           | CASIA1.0  | Columbia  | NIST16    | Coverage  | Avg.F1    |
> | --------------------------- | --------- | --------- | --------- | --------- | --------- |
> | RGB-aligned-to-noise (Ours) | **0.873** | **0.939** | **0.480** | **0.659** | **0.738** |
> | noise-aligned-to-RGB        | 0.867     | 0.922     | 0.475     | 0.596     | 0.715     |
>
> We retrained the model with different group convolution settings (2, 4, and 8 groups) in the DFCM module, and the results are shown below (pixel-F1). Our method, which uses depthwise separable convolution (number of groups = number of input channels), achieved the best performance.
>
> | Grouping number of DFCM | CASIA1.0  | Columbia  | NIST16    | Coverage  | Avg.F1    |
> | ----------------------- | --------- | --------- | --------- | --------- | --------- |
> | Channels_in (Ours)      | **0.873** | 0.939     | **0.480** | **0.659** | **0.738** |
> | 2                       | 0.849     | 0.942     | 0.458     | 0.611     | 0.715     |
> | 4                       | 0.861     | 0.949     | 0.438     | 0.600     | 0.712     |
> | 8                       | 0.865     | **0.957** | 0.434     | 0.595     | 0.713     |
>
> Q3: It is hard to clearly understand the key mechanisms from Fig.2.
>
> A: We have revised Figure 2 by adding distinct color blocks to highlight the core mechanisms, optimized the layout to reduce visual clutter, and included textual annotations to improve comprehension.
>
> We sincerely appreciate your comments and hope that our explanations and extra experiment results can effectively address your concerns. We would be grateful if you could consider raising the scores.
>
> [1] Qu, Chenfan, et al. "Towards modern image manipulation localization: A large-scale dataset and novel methods." *Proceedings of the IEEE/CVF Conference on Computer Vision and Pattern Recognition*. 2024.
>
> [2] Su, Lei, et al. "Can we get rid of handcrafted feature extractors? sparsevit: Nonsemantics-centered, parameter-efficient image manipulation localization through spare-coding transformer." *Proceedings of the AAAI Conference on Artificial Intelligence*. Vol. 39. No. 7. 2025.
>
> [3] Zhang, Quan, et al. "IMDPrompter: Adapting SAM to image manipulation detection by cross-view automated prompt learning." *arXiv preprint arXiv:2502.02454* (2025).
>
> [4] Xu, Zhipei, et al. "Fakeshield: Explainable image forgery detection and localization via multi-modal large language models." *arXiv preprint arXiv:2410.02761* (2024).

---

### Official Review · Reviewer_4Eo6 · 2025-11-01

**Soundness:** 2
**Presentation:** 2
**Contribution:** 2
**Rating:** 4
**Confidence:** 4

**Summary:**

This paper proposes QMA-Net, an image manipulation localization (IML) framework that alternates feature extraction and cross-modal interaction between RGB and noise modalities, combined with a query-driven multi-level feature decoder for adaptive fusion. Experiments on several benchmarks show modest but consistent improvements over prior SoTA methods.

**Strengths:**

- According to reported results, the framework outperforms state-of-the-art methods across benchmarks.
- From a design perspective, the idea of alternating between extraction and interaction (rather than simply branching + fusing) is intuitive and potentially beneficial.

**Weaknesses:**

## Main Weakness
- The approach is fundamentally a refinement of the dual-branch + fusion paradigm. The “alternating” schedule, while reasonable, does not constitute a paradigm shift. The query-based decoding has been used in other domains (e.g., DETR, Visual Prompt Tuning, Mask2Former) which weakens the novelty claim.
- There is heavy reuse of ideas such as multi-scale or multi-modal feature extraction and fusion in existing literature. The paper does not convincingly show clear dissimilarity to or break from those approaches like MVSS or Mesorch.
- The authors do not provide the model’s complexity, and the frequent feature fusion raises concerns that it may be a network with very large parameter count and FLOPs.
- The manuscript contains many places where it does not meet mature academic writing standards. For example, in Figure 1 it is not self-contained: the term ‘HF’ in ‘OursHF’ is undefined, and the word ‘classic’ is used without clarifying which specific model is meant. In line 53 the text refers to ‘(columns 5 and 6)’ but earlier no clear reference is given to what image or table those columns correspond to. There are many similar instances throughout the paper. From the standpoint of writing and presentation, this falls short of the level expected at conferences like ICLR.

**Questions:**

Please reference to Weakness section.

---

> ### Author Response · Authors · 2025-11-21
> **Response to Reviewer 4Eo6**
>
> # Response to Reviewer 4Eo6
>
> Thank you for your comments. We will respond to your questions point by point.
>
> Q1: The approach is fundamentally a refinement of the dual-branch + fusion paradigm. The “alternating” schedule, while reasonable, does not constitute a paradigm shift. The query-based decoding has been used in other domains (e.g., DETR, Visual Prompt Tuning, Mask2Former) which weakens the novelty claim.
>
> A: We agree that both dual-branch architectures and query-based decoders have been explored in other vision tasks such as detection and segmentation. However, our contribution does not lie in simply combining these components; instead, the novelty stems from how feature extraction, cross-modal interaction, and query-driven decoding are organized, scheduled, and tightly coupled for the specific challenges of image manipulation localization (IML). We clarify the differences as follows:
>
> Most existing dual-branch IML models follow a static “extract → fuse” pipeline, where RGB and noise features are fused only at mid- or late-stage, and the fusion does not influence subsequent feature extraction. Moreover, these frameworks do not perform feature interaction. In contrast, our design introduces a multi-stage alternating feature extraction and interaction mechanism, where:
>
> - RGB and noise branches interact at every stage rather than only once;
>
> - The interaction results are fed back into both backbones before the next-stage extraction;
>
> - Cross-modal guidance becomes iterative, progressive, and accumulative, not one-shot.
>
> Our ablation studies and CAM analysis also demonstrate that our method constructs a tampering feature representation that is more sensitive to artifacts and achieves superior performance compared to traditional paradigms.
>
> Both our Query-Driven Decoding and Mask-to-Former utilize the concept of a "query." However, our query design is not a direct adaptation of these works:
>
> - Our entire Query-Driven Decoding pipeline is entirely different from these methods.
>
> - Queries are updated progressively across multi-level MFAMs, each conditioned on the cross-modal interaction results from the alternating stages.
>
> - Queries in our framework serve as the bridge that links the alternating backbone interaction with the final mask generation, rather than being an independent decoding layer.
>
> - Our queries are tamper-aware forensic queries, trained to selectively aggregate manipulation-related features rather than object categories or semantic masks.
>
>
>
>
>
> Q2: There is heavy reuse of ideas such as multi-scale or multi-modal feature extraction and fusion in existing literature. The paper does not convincingly show clear dissimilarity to or break from those approaches like MVSS or Mesorch.
>
> A: Extensive experiments and research have demonstrated that the "multi-scale and multi-modal feature" are highly suitable for image tampering localization tasks, which is why we have adopted it. However, our innovations primarily focus on the overall framework and components—specifically, the novel interaction paradigm and the query-driven decoding strategy.
>
> The key distinction from MVSS/Mesorch lies in:
>
> MVSS does not employ multi-scale features at all. Both MVSS and Mesorch follow the classic "feature extraction-fusion" paradigm but lack the process of feature interaction. Thus, they are insufficient to capture the complementarity between the two modal features. Moreover, their fusion is only performed at the final stage. Our method employs an alternating architecture for feature extraction and interaction. Through multi-stage, progressive global interaction, we successfully construct a complementary tampering feature representation. This architecture has been demonstrated in the paper to be superior to traditional paradigms.
>
> MVSS and Mesorch employ overly simplistic decoding strategies that fail to account for interference from irrelevant features while introducing excessive additional parameters. In contrast, our query-driven decoding follows a progressive approach that continuously aggregates critical information from multi-level features throughout the entire framework, effectively serving as a feature filtering mechanism.
>
> Our comparative experiments also demonstrate that our method markedly outperforms MVSS (+40.8%) and Mesorch (+10.6%).

---

> ### Author Response · Authors · 2025-11-21
> **Response to Reviewer 4Eo6**
>
> # Response to Reviewer 4Eo6
> Q3: The authors do not provide the model’s complexity, and the frequent feature fusion raises concerns that it may be a network with very large parameter count and FLOPs.
>
> A: We report the parameter counts and FLOPs of our model and other baselines in Appendix 8. The results are as follows:
>
> | Method         | Parameters (M) | FLOPS (G) |
> | -------------- | -------------- | --------- |
> | ManTra-Net     | 3.9            | 274.0     |
> | MVSS-Net       | 150.5          | 171.0     |
> | PSCC-Net       | 3.7            | 376.8     |
> | CAT-Net        | 116.7          | 137.2     |
> | TruFor         | 68.7           | 236.2     |
> | SAM            | 309.0          | 1499.0    |
> | IMDPrompter    | 347.6          | 1533.0    |
> | QMA-Net (Ours) | 114.0          | 230.0     |
>
> It can be seen that the parameter count and FLOPs of our method are at a relatively low level compared to other baseline models.
>
> Q4: The manuscript contains many places where it does not meet mature academic writing standards. For example, in Figure 1 it is not self-contained: the term ‘HF’ in ‘OursHF’ is undefined, and the word ‘classic’ is used without clarifying which specific model is meant. In line 53 the text refers to ‘(columns 5 and 6)’ but earlier no clear reference is given to what image or table those columns correspond to. There are many similar instances throughout the paper. From the standpoint of writing and presentation, this falls short of the level expected at conferences like ICLR.
>
> A: Thank you for pointing out these writing typos. We have thoroughly reviewed the entire paper and corrected all typos.
>
> We sincerely appreciate your comments and hope that our explanations and extra experiment results can effectively address your concerns. We would be grateful if you could consider raising the scores.

---

### Official Review · Reviewer_dkTH · 2025-11-01

**Soundness:** 4
**Presentation:** 4
**Contribution:** 3
**Rating:** 8
**Confidence:** 5

**Summary:**

This paper proposes a novel framework named QMA-Net, which primarily comprises two key components: a Multi-stage Alternating Feature Extraction and Interaction architecture and lightweight, Query-driven Multi-level Feature Decoding. At each stage, the intrinsic relationships and mappings between RGB modality and noise modality features are deeply explored through the Cross-modal Feature Alignment and the DFCM. Simultaneously, the most critical information is selectively extracted and condensed from the dual-modality features while filtering out irrelevant interference via the MFAM and Tamper-Aware Queries. The experimental results demonstrate that QMA-Net outperforms existing state-of-the-art methods in both localization accuracy and robustness.

**Strengths:**

1. The proposed Multi-stage Alternating Feature Extraction and Interaction architecture constructs complementary bimodal feature representations, thereby effectively enhancing the sensitivity to tampering artifacts.
2. The unique decoding mechanism effectively focuses on key regions while filtering out irrelevant interference.
3. The writing is easy to understand.
4. To enhance interpretability, the authors visualized features from different levels. An extensive set of ablation studies was conducted, which thoroughly validates the model's design.
5. The overall model has fewer parameters and lower computational requirements.

**Weaknesses:**

1. There is no analysis of failure cases, which limits transparency about when and why the model may fail in real-world forensic scenarios.
2. In Figure 2's QMA-Net pipeline, the notations (e.g., R1 through R4) are somewhat compact. The overall layout could be made more spacious.
3. Fail to compare with more effective models such as APSC-Net [1], or discuss them in related works, limiting the demonstrated effectiveness.
4. There are a few typos in the paper.

[1] Qu C, Zhong Y, Liu C, et al. Towards modern image manipulation localization: A large-scale dataset and novel methods[C]//Proceedings of the IEEE/CVF Conference on Computer Vision and Pattern Recognition. 2024: 10781-10790.

**Questions:**

none

---

> ### Author Response · Authors · 2025-11-21
> **Response to Reviewer dkTH**
>
> # Response to Reviewer dkTH
>
> Thank you for your comments. We will respond to your questions point by point.
>
> Q1: There is no analysis of failure cases, which limits transparency about when and why the model may fail in real-world forensic scenarios.
>
> A: We analyzed the tampered images in the test set that performed poorly in predictions and identified two main types of failure cases: small tampered regions and high-resolution images. For the former, we believe the possible reason is that small tampered areas contain fewer artifacts, and edge supervision is less effective. For the latter, we suspect that irrelevant artifacts introduced during the resizing process interfered with the judgment. These are all areas we aim to improve in the future.
>
> Q2: In Figure 2's QMA-Net pipeline, the notations (e.g., R1 through R4) are somewhat compact. The overall layout could be made more spacious.
>
> A: Due to space constraints, the images were highly compressed. In the latest version, we will present the images in their original dimensions and improve the layout of the symbols within them to better highlight our new mechanism.
>
> Q3: Fail to compare with more effective models such as APSC-Net [1], or discuss them in related works, limiting the demonstrated effectiveness.
>
> A: We retrained APSC-Net[1] and SparseViT[2] using the same training settings, and the test results are as follows (pixel-F1). The results for IMDPrompter[3] and FakeShield[4] are cited from the original paper.
>
> | Method         | CASIA1.0  | Columbia  | NIST16    | Coverage  | Avg.F1    |
> | -------------- | --------- | --------- | --------- | --------- | --------- |
> | APSC-Net       | 0.798     | 0.941     | **0.500** | 0.402     | 0.660     |
> | SparseViT      | 0.827     | **0.959** | 0.384     | 0.513     | 0.671     |
> | IMDPrompter    | 0.763     | 0.873     | 0.411     | 0.636     | 0.671     |
> | FakeShield     | 0.600     | 0.750     | 0.370     | \         | 0.573     |
> | QMA-Net (Ours) | **0.873** | 0.939     | 0.480     | **0.659** | **0.738** |
>
> The results indicate that our method significantly outperforms these models in terms of average performance.
>
> Q4: There are a few typos in the paper.
>
> A: We have thoroughly reviewed the entire paper and corrected all writing typos.
>
> We sincerely appreciate your comments and hope that our explanations and extra experiment results can effectively address your concerns. We would be grateful if you could consider raising the scores.
>
> [1] Qu, Chenfan, et al. "Towards modern image manipulation localization: A large-scale dataset and novel methods." *Proceedings of the IEEE/CVF Conference on Computer Vision and Pattern Recognition*. 2024.
>
> [2] Su, Lei, et al. "Can we get rid of handcrafted feature extractors? sparsevit: Nonsemantics-centered, parameter-efficient image manipulation localization through spare-coding transformer." *Proceedings of the AAAI Conference on Artificial Intelligence*. Vol. 39. No. 7. 2025.
>
> [3] Zhang, Quan, et al. "IMDPrompter: Adapting SAM to image manipulation detection by cross-view automated prompt learning." *arXiv preprint arXiv:2502.02454* (2025).
>
> [4] Xu, Zhipei, et al. "Fakeshield: Explainable image forgery detection and localization via multi-modal large language models." *arXiv preprint arXiv:2410.02761* (2024).

---

> > ### Comment · Reviewer_dkTH · 2025-11-23
> > **Response to Author Rebuttal**
> >
> > Thank you for your response, please add the new experiment results to table1, add submit the updated paper before the end of the rebuttal stage.

---

### Meta-Review · Area_Chair_vPQC · 2026-01-06

**Summary:**

This paper did not receive very strong support pre rebuttal.

The AC thinks that the paper mainly fails to motivate why the fairly complex earlier fusion mechanism helps in the realm of image tampering detection, instead relying heavily on empirical observations.

During rebuttal, the authors did present new results to support the validity of their early fusion method for tamper detection, but the WHY remains elusive. The failure cases identified during rebuttal are also concerning as the field of image tamper detection has not see many prior arts that have practical impact, and the authors only identified small tampered regions as the only failure?

Authors did add more experiments comparing to SOTA during the rebuttal.

The writing can be largely improved as many aspects were not properly articulated, particularly with respect to the WHY.

Weighing the pros and cons, the AC does not have confidence that this work will have practical impact.

**Reviewer Concerns:**

Rebuttal added more comparisons to SOTA. However, the method remains hard to understand why it helps to detect tampering.

**Reviewer Scores:**

AC does not think that the reviewer scores would have improved much from the rebuttal to warrant an accept.

---

### Decision · Program_Chairs · 2026-01-26

Reject